# Associations Between Candida and *Staphylococcus aureus, Pseudomonas aeruginosa*, and *Acinetobacter* Species as Ventilator-Associated Pneumonia Isolates in 84 Cohorts of ICU Patients

**DOI:** 10.3390/microorganisms13061181

**Published:** 2025-05-22

**Authors:** James Hurley

**Affiliations:** 1Melbourne Medical School, University of Melbourne, Parkville, VIC 3052, Australia; hurleyjc@unimelb.edu.au; 2Ballarat Health Services, Grampians Health, Ballarat, VIC 3350, Australia; 3Ballarat Clinical School, Deakin University, Ballarat, VIC 3350, Australia

**Keywords:** ventilator-associated pneumonia, intensive care unit, observational cohort, mechanical ventilation, *Staphylococcus aureus*, *Pseudomonas aeruginosa*, *Acinetobacter* species, *Candida* species

## Abstract

*Staphylococcus aureus, Pseudomonas aeruginosa*, *Acinetobacter* species, and *Candida* species are common ventilator-associated pneumonia (VAP) isolates. Whilst the clinical significance of *Candida* as a VAP isolate is unclear, evidence is emerging that *Candida* interacts with bacteria, contributing to colonization susceptibility. Indirectly, VAP isolate data reflect patient colonization within cohorts. The objective here is to estimate the association between these three bacteria and Candida as VAP isolates. ICU cohorts were obtained by searching the literature for mechanically ventilated (MV) patient cohorts in which Candida was listed as an isolate among patients with VAP. Regression models of the associated VAP incidence per 100 MV patients, using random effects methods, incorporated group-level factors such as the year of publication, mode of VAP diagnosis, and ICU stay length. The median VAP incidence proportions for *Staphylococcus aureus, Pseudomonas aeruginosa,* and *Acinetobacter* species were 3.3 (IQR: 1.2–6.9), 3.6 (IQR: 1.8–5.7), and 1.2 (IQR: 0.4–4.1), respectively. Among 84 cohorts from 67 publications, *Staphylococcus aureus* (correlation coefficient = 0.759) and *Pseudomonas aeruginosa* (0.749), and less so *Acinetobacter* species (0.53)*,* each show correlation with the isolation of *Candida* species among these ICU populations. These associations may underlie the poor prognosis with Candida colonization.

## 1. Introduction

Ventilator-associated Pneumonia (VAP) occurs in 5 to 40% of patients undergoing mechanical ventilation in intensive care units (ICUs). Common VAP isolates include *Staphylococcus aureus*, *Pseudomonas aeruginosa*, *Acinetobacter* species, and *Candida* species [1,2,3,4,5,6,7,8,9,10,11,12,13,14,15,16,17,18,19,20,21,22,23,24,25,26,27,28,29,30,31,32,33,34,35,36,37,38,39,40,41,42,43,44,45,46,47,48,49,50,51,52,53,54,55,56,57,58,59,60,61,62,63,64,65,66,67]. The clinical relevance of *Candida* species as a VAP isolate remains uncertain, and true Candida pneumonia in this patient population is considered rare. A review of 2490 isolates from 24 studies revealed that fungi (the species were not specified) constituted only 0.9% of pathogenic VAP isolates [68]. Current guidelines [69,70] do not advocate for the treatment of Candida isolated from the respiratory tract of patients with VAP in the absence of specific evidence of infection.

Candida colonization is associated with poorer outcomes among ICU patients [71,72,73,74,75]. Surprisingly, this may not relate to invasive Candida infection, as a patient’s history of Candida colonization has a low positive predictive value for ruling in Candida as a potential cause of sepsis [73]. On the other hand, there is great interest in whether Candida colonization mediates colonization susceptibility, wherein it facilitates the ability of bacterial pathogens to cause VAP, along with other clinically important infections [76,77,78,79,80,81,82,83,84,85,86,87,88]. This process, termed ‘microbial hitchhiking’ [80], would have population-level implications.

Candida colonization is a difficult endpoint to study. It is variably defined, measured, and reported in the studies that have been published on this topic. On the other hand, Candida as a VAP isolate, which is better defined, can serve as a proxy indicator of Candida colonization among the ICU patient cohort.

Whilst the mechanisms that might underlie colonization susceptibility mediated by *Candida* have been studied experimentally in numerous and extensive models, any clinical significance remains difficult to determine. Various approaches have been used in attempts to better define its clinical significance [78,79,80,81,82,83,84,85,86,87,88].

The interactions that might underlie these interactions between bacteria and fungi have been widely studied with respect to a range of microbes and also to a range of potentially relevant sites in the body, such as the intestinal tract, wounds, the skin and the vagina. Other studies have examined the impact of the presence of Candida within polymicrobial experimental infections.

Here, the interest is specifically in the associations between the three bacteria and *Candida* species as VAP isolates that might be relevant to causing infections of the respiratory tract. The objective here is to define the extent of association between each of *Staphylococcus aureus, Pseudomonas aeruginosa*, and *Acinetobacter* species with *Candida* species as VAP isolates, based on published series that list *Candida* species among VAP isolates.

## 2. Materials and Methods

Being an analysis of published work, ethics committee review of this study was not required.

### 2.1. Study Selection and Grouping of Cohorts

The literature search used here is as described previously [77,84,85,86,87,88]. In brief, a search was undertaken using the following terms: “ventilator-associated pneumonia”, “mechanical ventilation”, and “intensive care unit”, each combined with either “meta-analysis” or “systematic review”, up to December 2024. Cochrane reviews and other systematic reviews were used as the primary source of studies. Additional studies were identified through snowball sampling of the literature using the “Related articles” function in Google Scholar. Any duplicate studies were removed. Studies in a language other than English were included when the required data had been abstracted from an English-language systematic review.

The inclusion criteria were cohorts of ICU patients for whom >90%, an arbitrary threshold, received prolonged (>24 h) mechanical ventilation, for which *Candida* species were listed among the VAP isolates. Where possible, data were extracted for each identifiable sub-cohort representing different patient types, age groups, or observation eras from the studies. Studies limited to patients with the acute respiratory distress syndrome (ARDS) and pediatric cohorts were not excluded.

Studies of antimicrobial interventions used to prevent ICU-acquired infections were excluded, as these would be expected to alter the correlation between the three bacteria and Candida. Studies with fewer than 50 patients were also excluded.

### 2.2. Outcomes of Interest

The independent variable in the regression models was the count of VAP patients with Candida listed among the respiratory tract (RT) isolates as a proportion of the total number of patients receiving mechanical ventilation. The dependent variables were the counts of VAP patients with each of the *Staphylococcus aureus*, *Pseudomonas aeruginosa*, or *Acinetobacter* species listed among the VAP isolates. These counts were extracted as proportions of the total number of patients receiving mechanical ventilation in each publication.

A range of other endpoints of interest were extracted from the publications. These included the year of study publication, the group mean or median LOS, or the group mean or median duration of mechanical ventilation, whether the method of obtaining VAP isolates was by bronchoscopic or non-bronchoscopic methods, and the geographic region of origin of the ICU.

The VAP isolate data, being count data, were logit transformed using the number of patients receiving prolonged (>24 h) mechanical ventilation as the denominator. Note that a logit transformation requires the addition of 0.5 as a continuity correction to any zero counts to enable zero-event groups to be represented on the logit scale, as the logit transformation of zero is indeterminate. The LOS data were positively skewed and were transformed for all analyses as follows: any LOS < 4 days was truncated to 4 days; the LOS was divided by 7 and then log transformed.

### 2.3. Regression Model

Models of the relationship between VAP isolates for each of the three categories of bacterial isolates were generated using regression analysis with the glm command in Stata 18 (Stata 18, College Station, TX, USA). The marginal effects associated with increasing counts of RT candida were obtained using the margins command following each glm model.

### 2.4. Sensitivity Tests

VAP associated with *Acinetobacter* species occurs in approximately 1% of patients in some cohorts. Given the limited nature of this count, the analysis was initially undertaken on groups with >100 patients, and then repeated with the inclusion of groups with <100 patients as a sensitivity test.

### 2.5. Missing Data

Cohorts that lacked the number of patients receiving mechanical ventilation were incorporated by imputing a denominator on the assumption that the VAP patients represent approximately 20% of patients receiving mechanical ventilation in typical cohorts. The analysis was repeated with and without the inclusion of these studies. The missing data in this study were determined to be missing at random. The percentage of missing VAP isolate count data was generally less than 10%. To mitigate bias from the exclusion of observations due to missing data, multiple imputation was employed by imputing at least 10 datasets for analysis [89]. The mode of patient diagnostic sampling, whether bronchoscopic or non-bronchoscopic, was analyzed using the available data without imputation (<4% missing).

### 2.6. Bivariate Plots and Confidence Ellipses

To assess the correlation between RT Candida and candidemia incidence, the logit-transformed data were analyzed using a 95% prediction ellipse in addition to linear regression. The prediction ellipse method on a logit scale is optimal for enabling the bivariate correlation to be observed. The confidence ellipses were generated using the ‘ellip’ command within Stata [90].

## 3. Results

### 3.1. Characteristics of the Studies

There were 84 cohorts derived from 67 published studies identified by the search (Table 1).

The studies were published between 1982 and 2020, with the interquartile range (IQR) being 1998–2010. The median ICU LOS was 10 days (IQR: 6.4–14.4), and the median cohort size was 410 patients (IQR: 124–1514). There were twelve cohorts with fewer than 100 patients. Fourteen studies provided more than one observational group. Most studies originated from ICUs in Northern Europe (*n* = 24), Southern Europe (*n* = 13), or North America (*n* = 11), and five were multinational. A minority were cohorts of trauma patients, and for the remaining cohorts, the median proportion of trauma patients was 5 (IQR: 1–20). There were no pediatric cohorts.

VAP isolates were obtained by bronchoscopic sampling in 26 studies and by non-bronchoscopic methods in the other 41.

The median VAP incidence proportion was 16.3% (IQR: 7.5–21%). Eight cohorts required the imputation of denominator data, being the number of patients receiving mechanical ventilation.

The median VAP incidence proportions for *Staphylococcus aureus, Pseudomonas aeruginosa,* and *Acinetobacter* species were 3.3% (IQR: 1.2–6.9), 3.6% (IQR: 1.8–5.7), and 1.2% (IQR: 0.4–4.1), respectively. The proportion of patients with Candida as a respiratory tract isolate in association with VAP was 0.5% (IQR: 0.13–1.5).

### 3.2. Scatterplots

The scatterplots of isolate counts and the accompanying ellipse plots are presented in Figure 1, Figure 2 and Figure 3. *Staphylococcus aureus* (correlation coefficient = 0.749) and *Pseudomonas aeruginosa* (0.722), and to a lesser extent, *Acinetobacter* species (0.535), each show a correlation with the isolation of *Candida* species.

### 3.3. Regression Models and Sensitivity Tests

The regression models are presented in Table 2 for models based on either the original data or with the imputation of missing data. The Candida count was a significant correlate of the counts of each of the three bacteria, even after adjusting for group mean ICU LOS, year of study publication, and mode of diagnostic sampling. The group mean LOS was a factor of marginal statistical significance. The results of all models were robust to the accommodation of missing VAP isolate count data using multiple imputation. The results of all models were robust to the inclusion of eight cohorts with imputed denominator data as described in the Materials and Methods, and the associated coefficients were similar. The results with the inclusion of twelve cohorts with <100 patients were similar. The results limited to cohorts published since 2000 were similar.

The marginal effect associated with the difference between an RT Candida of 1 and 2 percent (a 1 percentage point increase) for each of the three bacteria was an approximately 1.4 percentage point increase.

## 4. Discussion

*Staphylococcus aureus*, *Pseudomonas aeruginosa*, and *Acinetobacter* species, as well as *Candida* species, are common VAP isolates. This study presents an analysis to define the extent of association between *Staphylococcus aureus*, *Pseudomonas aeruginosa*, and *Acinetobacter* species, and *Candida* species, as VAP isolates in published series that list *Candida* species among isolates from patients with VAP. The count of Candida among VAP isolate data serves as a proxy measure of Candida colonization within these ICU patient cohorts.

Whilst the clinical significance of *Candida* colonization is uncertain, the possibility that this microorganism might mediate colonization susceptibility among ICU patient cohorts is of great interest. A growing interest is emerging in the respiratory microbiome of ICU patients and its relationship to therapeutic interventions [91,92,93,94].

Interactions between fungi and bacteria can be classified into at least four broad categories [79,80,81,82]. Direct binding of bacteria by Candida potentially facilitates intestinal translocation into the host [95]. In confined environments, the release or consumption of chemical compounds, such as metabolic byproducts or quorum-sensing molecules, underlies communication between bacteria and fungi. Biochemical changes can occur due to either the consumption of oxygen or alterations in pH that result from proton release. Within the host, both specific and non-specific immune system responses may be induced by the presence of Candida.

Some of these interactions may result in effects that might be either synergistic or antagonistic to any bacteria present. For example, in a rodent model of pneumonia, it has been observed that an inoculum of *P. aeruginosa* in the respiratory tract, which alone is insufficient to cause bacterial pneumonia, was able to induce infection when co-instilled with *C. albicans* [96]. This effect was not seen when ethanol-killed *C. albicans* was used, indicating that the Candida must be viable in order to enhance the virulence of *P. aeruginosa*. This phenomenon has been consistently observed across challenge infections with various bacterial pathogens, including *Staphylococcus aureus* and *Escherichia coli* [97].

While animal models permit the manipulation of microbial components within the microbiome, this is not possible in human studies, which are primarily correlative [56,78,83,98].

There are several technical challenges in studying the significance of any interactions between fungi and bacteria. For instance, although many experiments utilize mice as the host organism, it is important to note that *Candida albicans* is not typically a commensal organism in many laboratory mice [99,100,101]. Nonetheless, prolonged colonization can be induced under specific conditions, such as exposure of mice to broad-spectrum antibiotics. Additionally, mice housed in proximity to those with antibiotic-induced colonization are at risk of cross-colonization with Candida from their counterparts [101]. This is another observation that relates to the population-level relevance of Candida colonization.

Another impediment to assessing the consequences of interactions among bacterial and fungal colonizing flora within patients is that there may be effects of Candida colonization beyond its mere presence among the colonizing flora. Many of the interaction effects mediated by Candida on bacteria, as defined in experimental settings, relate to the functional activity of Candida. It is not possible to gauge this functional activity within patients, as usually the mere presence or absence among the colonizing flora is reported. This presence typically does not convey the density of the Candida colonization.

There are several impediments to defining the possible clinical relevance of *Candida* colonization within the respiratory microbiome of patients receiving mechanical ventilation in the ICU [71,72,73,74,75,101,102,103]. First, colonization is variably defined in different studies depending on the methods, site, and timing of its assessment. By using the listing among the VAP isolates of each of Candida and the three potentially associated bacteria, some comparability is achieved across the studies included here. Second, there is a commonality among the important risk factors for colonization with the organisms under study here [104,105]. For example, a major risk factor for acquiring colonization with *Staphylococcus aureus*, Enterococcus, Gram-negative bacteria, *Clostridium difficile*, and Candida is the population-level use of antimicrobials within the ICU. Other risk factors that are difficult to measure include the risk of cross-infection, staffing factors, and the colonization density of various microbes within the ICU, which is determined by the number of patients who are colonized.

### 4.1. Limitations

There are several limitations to this analysis. The first limitation to be considered is the question of non-reporting of Candida amongst potentially eligible studies. This limitation was addressed by limiting the inclusion to those studies that had reported a count of Candida among the respiratory-tract isolates of patients with VAP. However, the exclusion of studies that failed to report this count creates a potential bias.

Second, this is a group-level analysis, and the inferences from the regression models may not correspond to inferences at the patient level. In particular, LOS in the ICU is a strong correlate of colonization with a range of pathogens. The estimates here are derived using group-level mean LOS values, which would likely underestimate the strength of this factor as a patient-level correlate of colonization. Moreover, the simple use of group mean LOS may obscure more complex non-linear relationships with the emergence of different patient infections in various ICU populations. For example, evidence suggests that head injury and various forms of trauma may increase the risk of *Acinetobacter* species. However, these risk factors may be confounded by the length of stay (LOS) in healthcare settings, and more specifically, the duration of mechanical ventilation prior to the onset of VAP [106].

Third, the ICU populations, which have been published over at least four decades, have been broadly selected and are heterogeneous. This may be both a strength and a limitation, as the findings are not specific to any single population but are broadly representative of the literature experience. The year of study publication will only approximate when the cohort of ICU patients was actually studied. This is unlikely to be a major concern given the four-decade time period over which the studies were conducted. Additionally, the year of study publication was not a significant factor in any of the models for any of the three bacteria.

Fourth, the published studies had missing data for various endpoints, including the counts of bacteria, which were the endpoints of interest. Repeating the analysis with imputed data for the missing data gave similar findings. The fact that bronchoscopic sampling was not a significant factor in the analysis may reflect the level of analysis, which was not at the individual patient data level. Whilst individual patient data would have provided greater precision in the analysis, it was not available for studies that were mostly published several decades ago.

Fifth, even with 84 cohorts from 67 published studies, the amount of data is not sufficient to fully explore some of the potentially relevant factors. For example, the number of cohorts in this study was underpowered, and the method may not have been optimal for demonstrating regional and temporal trends that were previously apparent in an analysis with a larger number of more broadly selected studies [84,85,86,87,88]. Moreover, the analysis is limited to VAP isolates. On the other hand, colonization susceptibility may have more relevance to invasive infections such as bacteremia [102]. However, bloodstream infection, being a much rarer endpoint, is more difficult to study [107,108].

Some of the included studies were small in size, and this may have been a limiting factor in observing rare outcomes. Also, there is the potential for the types of VAP isolates to have changed over the four decades that the included studies span. These two limitations were addressed by excluding studies with fewer than 100 patients and by limiting the analyses to those studies published since the year 2000, respectively. These analyses did not change the findings.

Of note, studies on different antimicrobial methods as prophylaxis for preventing bacterial infections arising from colonization in the ICU were specifically excluded from this analysis. Whilst these would add statistical power, they would be expected to have an altered correlation between the three bacteria and Candida. Their inclusion would obscure the assessment of the natural correlation between the three bacteria and the isolation of *Candida* species within populations of ICU patients under conditions of usual care, which is the objective here.

The sixth limitation is that this review has not considered the interactions that bacteria might have with other bacteria. For example, the α-toxin released by *S. aureus*, which is a cytotoxic agent, contributes to the proliferation and survival of *A. baumannii* in vitro [109].

Finally, this study has not examined outcomes among patients who received various strategies of decolonization using antifungal agents as prophylaxis to control Candida colonization. The interpretation of these studies is complex, and their findings are mixed. For example, one cohort study compared patients who were mechanically ventilated for over two days and treated with nebulized amphotericin B for Candida spp. airway decolonization with a propensity-matched control group who did not receive the decolonization treatment [110]. Nebulized amphotericin B was associated with a lower risk of *P. aeruginosa* VAP and improved 90-day mortality rates among critically ill patients with Candida spp. tracheobronchial colonization. In contrast, however, this improvement in outcomes was not corroborated by a recent systematic review of nine studies investigating antifungal prophylaxis in this patient population [111]. On the other hand, a study in a Greek ICU found higher mortality among patients receiving antifungal therapies [112].

### 4.2. Comparisons with Other Findings

Kollef et al. prospectively surveyed *P. aeruginosa* VAP among 1873 mechanically ventilated patients in 56 ICUs from 11 countries in Europe, the United States, Latin America, and the Asia-Pacific region [113]. The incidence of VAP overall and *P. aeruginosa* VAP among all 56 ICUs was 293/1873 (15.6%) and 76/1873 (4.1%), respectively, which compares to incidence proportions for *P. aeruginosa* associated VAP found here.

In a trial of an intervention for preventing *Staphylococcus aureus* VAP, which prospectively recruited 767 mechanically ventilated patients in 31 ICUs from nine countries in Europe, and the Mediterranean region [114], the incidence of *Staphylococcus aureus* VAP was 43/767 (5.6%), which compares to incidence proportions for *Staphylococcus aureus* associated VAP found here.

The incidence of *Acinetobacter* VAP varies in different regions of the world, being higher in reports from the Middle East than reports from Northern Europe and North America [88]. A systematic review of 13 observational studies published from ICUs in Spain, Belgium, Taiwan, China, France, Lebanon, Portugal, Turkey, Bosnia and Herzegovina, the United States, and Nigeria since 2010 found that the incidence of VAP was 20.5% and *Acinetobacter baumannii* was the commonest cause of VAP episodes [115].

Two recent observational studies reported the incidence rates of COVID-related VAP. The first, involving 1576 mechanically ventilated patients in intensive care units (ICUs) across European countries [116], found the overall incidence of VAP to be 399/1576 (25.3%), with specific incidences for *Staphylococcus aureus* at 45/1576 (2.9%), *P. aeruginosa* at 92/1576 (5.8%), and *Acinetobacter* at 35/1576 (2.2%).

The second observational study of COVID-related VAP among 3388 mechanically ventilated patients in ICUs from European countries [117] revealed higher incidences. The overall incidence of VAP was 1523/3388 (45%), with *Staphylococcus aureus* accounting for 242/3388 (7.1%), *P. aeruginosa* for 286/3388 (8.4%), and *Acinetobacter* for 19/3388 (0.5%). The data from these two observational studies align with the incidence proportions observed in the present study.

Of note, neither of the COVID-related VAP cohorts provided information regarding the isolation of *Candida* species from VAP patients, and hence, are not included in the analysis here.

## 5. Population Relevance

There are various interventions employed to prevent the colonization of patients in intensive care units (ICUs) and thereby mitigate the risk of acquired infections. These are either antimicrobial-based or non-antimicrobial-based. It is crucial to consider whether antimicrobial-based interventions designed to target a specific bacterium may inadvertently spill over to adversely affect other bacteria within the ICU population [118,119,120]. This is especially relevant for colonization with Candida, which may not always be considered even within studies estimating the effect of these antimicrobial-based interventions on antimicrobial-resistant organisms [121]. Moreover, rebound colonization effects following the cessation or withdrawal of antimicrobial-based interventions have not been well studied.

It is important to consider the bacteria of interest at the level of the whole population, not merely patient subgroups within the ICU. For instance, a case-control study conducted in a 52-bed ICU of a university-affiliated hospital found that vancomycin-resistant enterococcus and methicillin-resistant *Staphylococcus aureus* commonly coexist within the colonizing flora of ICU patients and share risk factors. Similarly, gastrointestinal carriage of vancomycin-resistant enterococcus and carbapenem-resistant Gram-negative bacteria among hospitalized patients shared several common risk factors [122].

Also of note is the issue of co-infection and polymicrobial infections. For example, a subset of patients diagnosed with candidemia also presented with methicillin-resistant *Staphylococcus aureus* coinfection. This necessitates increased vigilance for polymicrobial infections arising from colonization following the identification of a primary bloodstream infection pathogen [123].

Other approaches are being utilized to identify various risk factors for co-colonization in different ward settings beyond the ICU. For example, Shapiro et al. employed a metapopulation framework to explain variations in the incidence of infections caused by seven major bacterial species and their drug-resistant variants across a network of 357 French hospital wards [124]. Their findings indicated that ward-level consumption of antibiotics such as piperacillin/tazobactam had a substantial impact on the incidence of more resistant pathogens, while connectivity (the transfer of patients between wards) significantly influenced hospital-endemic species and carbapenem-resistant pathogens.

There is growing interest in symbiotic fungi that colonize the respiratory tract and other areas, focusing on the impact of fungal dysbiosis on human health [125]. One recent method includes fecal microbiota transplantation to address colonization by problematic bacteria. Fecal microbiota transplantation seeks to restore the diversity and functionality of commensal microbiota, aiding recovery from dysbiosis linked to the presence of multidrug-resistant microorganisms and helping to reestablish colonization resistance to these pathogens. The concept of restoring healthy microbiota through fecal microbiota transplantation in critically ill patients is emerging as a promising approach to address dysbiosis in intensive care unit patient cohorts [126,127,128].

## 6. Conclusions

There is a range of technical challenges in studying population-level consequences of interactions among bacterial and fungal colonizing flora within patients receiving mechanical ventilation in the ICU setting. Among 84 cohorts from 67 publications, *Staphylococcus aureus* (correlation coefficient = 0.759), *Pseudomonas aeruginosa* (0.749), and, to a lesser extent, *Acinetobacter* species (0.53) each show a correlation with the isolation of *Candida* species among patients identified with VAP receiving prolonged mechanical ventilation in the ICU. This work contributes to our understanding of microbial interactions in VAP and may inform future interventional studies targeting fungal colonization to mitigate bacterial pneumonia risk.

## Figures and Tables

**Figure 1 microorganisms-13-01181-f001:**
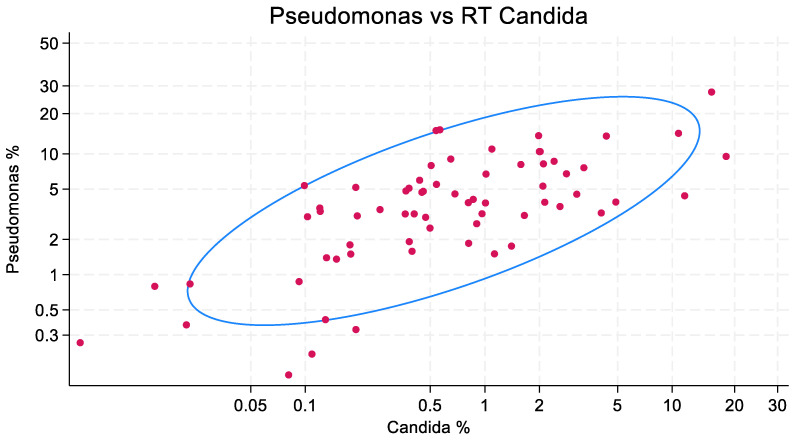
Correlation between the counts of patients with VAP caused by *Pseudomonas aeruginosa* and Candida among listed respiratory tract (RT) isolates, expressed as a percentage of all mechanically ventilated patients, in published cohorts (*n* = 68). Each red dot represents one cohort. The blue ellipse represents the mean-centered 95% confidence ellipse. Both axes are logit-scaled. The correlation coefficient is 0.749. RT candida proportions of 0.5, 1, and 2% were associated with corresponding increases in *P aeruginosa* proportions from 3.6 to 4.7 and 6.1, respectively.

**Figure 2 microorganisms-13-01181-f002:**
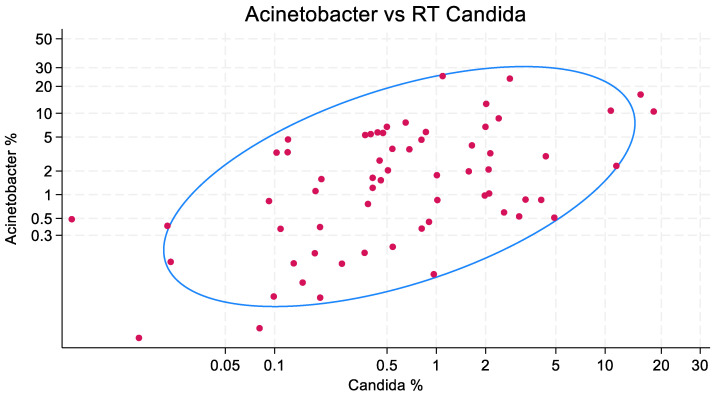
Correlation between the counts of patients with VAP caused by *Acinetobacter* species and Candida among listed RT isolates, expressed as a percentage of all mechanically ventilated patients, in published cohorts (*n* = 64). Each red dot represents one cohort. The blue ellipse represents the mean-centered 95% confidence ellipse. Note the axes are logit-scaled. The correlation coefficient is 0.53. RT candida at 0.5, 1, and 2% were associated with corresponding increases in *Acinetobacter* from 1.6 to 2.4 and 3.7, respectively.

**Figure 3 microorganisms-13-01181-f003:**
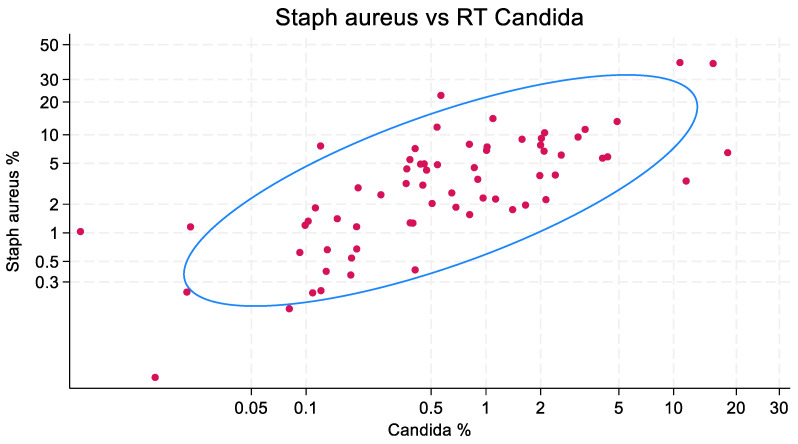
Correlation between the counts of patients with VAP caused by *Staphylococcus aureus* and Candida among listed RT isolates, expressed as a percentage of all mechanically ventilated patients, in published cohorts (*n* = 68). Each red dot represents one cohort. The blue ellipse represents the mean-centered 95% confidence ellipse. Note the axes are logit-scaled. The correlation coefficient is 0.759. RT candida at 0.5, 1, and 2% were associated with corresponding increases in *Staphylococcus aureus* from 2.8 to 4.0 and 5.7, respectively.

**Table 1 microorganisms-13-01181-t001:** Data extracted from studies listing Candida among reported VAP patient isolates.

				Patients	VAP Isolates (*n*)			
Author	Year	Ref.	LOS	(*n*)	Candida	Pseudomonas	Acinetobacter	Staph Aureus
Antonelli	1994	1	16.8	124	1	5	6	10
Bercault	2001	2	26	1144	2	·	·	·
Boots	2008	3	12.6	412	0	15	14	32
Bregeon	1997	4	11	660	3	33	10	34
Cade	1993	5	16	98	5	4	0	13
Cavalcanti	2006	6	10	190	6	9	1	18
Cenderero	1999	7	6.5	123	0	4	2	9
Chastre_ARDS	1998	8	25	56	6	8	6	22
Chastre_no_ARDS	1998	8	15	187	1	28	7	22
Chevret	1993	9	5	255	4	21	5	23
Cook_trauma	2010	10	8	511	1	16	8	15
Cook_non-trauma	2010	10	13	2080	4	7	1	14
Craven-surgical	1988	11	5.7	521	5	17	0	12
Craven-medical	1988	11	6.4	277	1	9	0	9
Daschner_Freiburg	1982	12	5	5374	7	22	·	21
Daschner_Switzerland	1982	12	5	1578	6	30	·	20
Daschner	1988	13	6	142	22	39	23	55
Daschner	1988	13	6	116	4	9	1	13
Delle Rose	2016	14	5	1647	27	52	68	32
Ewig	1999	15	10	48	1	4	0	5
Fagon’00_invasive	2000	16	10.4	^1^	5	27	6	20
Fagon’00_clinical	2000	16	10.7	^1^	38	57	11	40
Fowler	2003	17	15	^1^	2	28	3	30
George	1998	18	8	223	2	6	1	8
Giamarellos-Bourboulis	2009	19	12	72	2	5	17	·
Guérin	1997	20	18.5	260	0	14	1	3
Gursel	2010	21	10	92	1	10	23	13
Heyland	1999	22	7	1014	26	38	6	64
Holzapfel	1999	23	16	399	4	16	7	28
Hugonnet	2007	24	6	936	40	31	8	55
Ibrahim	2000	25	4.8	1882	19	130	16	143
Jimenez	1989	26	10	77	0	7	6	2
John	2017	27	6	202	1	5	14	·
Kantorova	2004	28	9	287	4	5	·	5
Kautzky	2014	29	20	65	2	·	·	·
Kollef ‘97_pre	1997	30	4	353	1	7	2	5
Kollef ‘97_post	1997	30	4	327	0	3	1	4
Kolpa	2018	31	19	1270	5	20	72	16
Koss– N	2001	32	14	87	10	4	2	3
Koss– P	2001	32	11	66	3	9	2	4
Laggner	1989	33	10.1	16	0	0	0	0
Li	2016	34	15	131	11	·	·	·
Luna	2003	35	8	427	2	13	25	19
Magnason	2008	36	7.8	280	0	5	0	1
Mahul	1992	37	21.5	145	3	8	3	10
Memish	2000	38	11	202	4	21	14	16
Moine	2002	39	14.4	764	2	27	1	19
Palabiyikoglu	2001	40	NA	50	1	·	·	·
Potgieter	1987	41	9	250	5	26	32	23
Ramirez	2016	42	13	440	0	·	·	8
Ranes	2006	43	NA	^1^	10	236	·	79
Rello’03	2003	44	20	99	0	8	2	2
Rello’91	1991	45	7.9	264	1	14	2	15
Rello’93	1993	46	NA	^1^	2	24	4	28
Resende	2013	47	22	126	3	11	11	5
Rodrigues	2009	48	10	233	2	10	14	11
Rosenthal	2012	49	6.9	3889	7	58	43	21
Rosenthal	2012	49	6.4	51,618	48	448	426	319
Rozaidi	2001	50	5	988	4	·	12	4
Ruiz-Santana	1987	51	7	1005	1	56	0	12
Salata	1987	52	11	51	1	7	0	2
Shahin	2013	53	10	267	3	4	·	6
Stolcin	2020	54	6	930	5	53	2	47
Sutherland_ARDS	1995	55	20	105	19	10	11	7
Terraneo	2016	56	22	^1^	21	69	1	65
Timsit	2001	57	30	^1^	2	21	4	10
Urli	2002	58	21	178	1	27	·	40
Verhamme	2007	59	8	4000	6	54	3	56
Xie	2011	60	7	4155	88	169	137	92
Kohlenberg	2010	61	NA	779,500	634	1067	152	1222
Lee	2013	62	NA	^1^	18	40	8	85
Leblebicioglu	2007	63	NA	3296	12	166	182	151
Leblebicioglu	2013	64	6	448	2	22	12	14
Leblebicioglu	2013	64	8	3864	4	119	130	51
Rosenthal_D	2006	65	5.3	3413	0	27	0	1
Rosenthal_C	2006	65	6.7	2172	0	18	3	25
Rosenthal_E	2006	65	6.3	1514	2	21	2	10
Rosenthal_H	2006	65	11.7	2305	10	142	137	118
Rosenthal_G	2006	65	5	1359	11	25	5	21
Rosenthal_B	2006	65	9.7	1029	7	49	38	19
Rosenthal_A	2006	65	5.5	8867	0	23	43	91
Rosenthal_F	2006	65	6.7	410	0	14	20	1
Tao	2011	66	NA	391,527	86	1446	1562	922
Tao	2012	67	NA	16,426	18	34	60	38

Abbreviations: LOS = Length of stay; NA = Not available; ‘·’ = count not reported; ^1^ = These cohorts lack denominator data but were included in the analysis as a sensitivity test after imputing denominator data, as described in the Materials and Methods.

**Table 2 microorganisms-13-01181-t002:** Regression models of isolate count data from cohorts with >100 patients.

		Original Data		Accounting for Missing Data with MI ^1^	
Model	Factor	Coefficient	95% Confidence Interval	Coefficient	95% Confidence Interval
*Staphylococcus aureus*	*Candida* ^2^	0.53	+0.37–+0.69	0.54	+0.38–+0.71
	LOS ^3^	0.04	−0.01–+0.08	0.04	−0.01–+0.08
	Year of study publication ^4^	−0.02	−0.05–+0.01	−0.02	−0.04–+0.01
	Bronchoscopic sampling ^5^	+0.14	−0.37–+0.66	+0.12	−0.39–+0.64
	Constant	−0.078	−1.88–+0.38	−0.071	−1.85–+0.42
*Pseudomonas aeruginosa*	*Candida* ^2^	0.42	+0.29–+0.54	0.42	+0.28–+0.56
	LOS ^3^	0.04	+0.01–+0.07	0.03	−0.01–+0.07
	Year of study publication ^4^	−0.02	−0.04–+0.01	−0.02	−0.04–+0.01
	Bronchoscopic sampling ^5^	−0.04	−0.44–+0.37	−0.05	−0.5–+0.40
	Constant	−1.09	−2.0–−0.22	−1.03	−2.0–−0.10
*Acinetobacter* species	*Candida* ^2^	0.57	+0.30–+0.83	0.59	+0.29–+0.88
	LOS ^3^	0.03	−0.05–+0.10	0.03	−0.06–+0.12
	Year of study publication ^4^	0.02	−0.03–+0.07	0.02	−0.04–+0.08
	Bronchoscopic sampling ^5^	−0.35	−1.23–+0.53	−0.36	−1.34–+0.62
	Constant	−2.17	−4.1–−0.26	−2.16	−4.53–−0.2

^1^ MI = multiple imputation methods used to impute missing data as described in the Materials and Methods; ^2^ = The proportion of patients with Candida among the VAP isolates; ^3^ = Per day of ICU LOS; ^4^ = Per year post 1980; ^5^ = Bronchoscopic versus non-bronchoscopic sampling.

## Data Availability

The data analyzed during the current study are provided in Table 1.

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
