# Peer review of "Associations Between Candida and Staphylococcus aureus, Pseudomonas aeruginosa, and Acinetobacter Species as Ventilator-Associated Pneumonia Isolates in 84 Cohorts of ICU Patients"

_microorganisms, 2025, doi:10.3390/microorganisms13061181_

Round 1
Reviewer 1 Report
Comments and Suggestions for Authors
This manuscript is lengthy and it is difficult to identify the novelty and significance of this study. Is the main result new?
Title:
The mention of “Population level” in the title is questionable given that 87 cohorts comprising tens to thousands patients were analysed.
Abstract
LL15-16. Here it should be explained clearly how “Candida interacts to enhance the pathogenic potential of bacteria causing colonization susceptibility”.
The number of cohorts should be checked: 67 cohorts appear in the title and 84 cohorts appear in Abstract and forth?
The key issue to be addressed in this study is difficult to understand from Introduction.
Why was Google Scholar, not PubMed, used to find additional studies?
Writing should be improved in many places and made understandable for broader audience.
For instance, the statement that “On the other hand, there is great interest in whether Candida colonization mediates colonization susceptibility wherein it facilitates the ability of bacterial pathogens to cause infection, a process termed ‘microbial hitchhiking” should be expressed more clearly. Candida colonization (where? on what?) colonization (of what?) susceptibility (of what?).
As can be understood from the Conclusion, the main result from this study is to highlight the occurrence /coexistence of Candida spp. in microbial populations isolated from the VAP patients. What is the novelty and significance of this study? Weren’t interactions between Candida and bacteria discussed in the literature?
The Discussion sections should be balanced with the produced results. How is this study relevant to discussing fungi-bacteria interactions? These interactions have been described too scarcely (LL 222-229) in contrast to available reviews. “Direct binding of bacteria by Candida” is mechanistic.
The strength of this study is that the limitations have comprehensively been considered in the separate subsection. Given the number of what justifies the use of this analysis?
Other comments
The text should be edited carefully.
L59: studies -> studied
eras -> ERAS (with spelling)
species should be typed as species (not in Italics)
Please avoid writing “gram-negative bacilli” in reference to rod-shaped bacteria. Representatives of the Bacillus genus (e.g. B. subtilis) are Gram-positive.
Comments on the Quality of English LanguageThe writing should be improved in many places, Some examples are shown in Comments.
Author Response
Comments 1: This manuscript is lengthy and it is difficult to identify the novelty and significance of this study. Is the main result new?
Response 1: comment added in manuscript conclusion statement that “The novelty here is that this work contributes to our understanding of microbial interactions in VAP and may suggest future interventional studies targeting fungal colonization to mitigate bacterial pneumonia risk.” And in abstract conclusion that “These associations may underlie the poor prognosis with Candida colonization.”
Title:
Comments 2: The mention of “Population level” in the title is questionable given that 87 cohorts comprising tens to thousands patients were analysed.
Response 2: the title edited for clarity with removal of ‘Population level’
Abstract
Comments 3: LL15-16. Here it should be explained clearly how “Candida interacts to enhance the pathogenic potential of bacteria causing colonization susceptibility”.
Response 3: See lines 43-49 of introduction [which have been clarified].
Comments 4: The number of cohorts should be checked: 67 cohorts appear in the title and 84 cohorts appear in Abstract and forth?
Response 4: abstract and title numbers reconciled.
Comments 5; The key issue to be addressed in this study is difficult to understand from Introduction.
Response 5: Abstract and introduction rewritten for clarity
Comments 6: Why was Google Scholar, not PubMed, used to find additional studies?
Response 5: This is not a systematic review. Cochrane reviews were used as the primary source.
Comments 7: Writing should be improved in many places and made understandable for broader audience.
For instance, the statement that “On the other hand, there is great interest in whether Candida colonization mediates colonization susceptibility wherein it facilitates the ability of bacterial pathogens to cause infection, a process termed ‘microbial hitchhiking” should be expressed more clearly. Candida colonization (where? on what?) colonization (of what?) susceptibility (of what?).
Response 7: lines 42-48 edited for clarity
Comments 8: As can be understood from the Conclusion, the main result from this study is to highlight the occurrence /coexistence of Candida spp. in microbial populations isolated from the VAP patients. What is the novelty and significance of this study? Weren’t interactions between Candida and bacteria discussed in the literature?
Response 8: comment added in manuscript conclusion statement that “The novelty here is that this work contributes to our understanding of microbial interactions in VAP and may suggest future interventional studies targeting fungal colonization to mitigate bacterial pneumonia risk.” And in abstract conclusion that “These associations may underlie the poor prognosis with Candida colonization.”
Comments 9: The Discussion sections should be balanced with the produced results. How is this study relevant to discussing fungi-bacteria interactions? These interactions have been described too scarcely (LL 222-229) in contrast to available reviews. “Direct binding of bacteria by Candida” is mechanistic.
Response 9: LL 222-229 edited for clarity. See also response: 8
Comments 10: The strength of this study is that the limitations have comprehensively been considered in the separate subsection. Given the number of what justifies the use of this analysis?
Response 10: Limitations edited for clarity
Other comments
Comments 11: The text should be edited carefully.
Response 11: The text has been carefully edited for clarity
Comments 12: L59: studies -> studied
Response 12: amended
Comments 13: eras -> ERAS (with spelling)
Response 13: amended
Comments 14: species should be typed as species (not in Italics)
Response 14: species typed as species (not in Italics)
Comments 15: Please avoid writing “gram-negative bacilli” in reference to rod-shaped bacteria. Representatives of the Bacillus genus (e.g. B. subtilis) are Gram-positive.
Response 15: These have been amended at each appearance
Reviewer 2 Report
Comments and Suggestions for Authors
Candida species—most notably Candida albicans—are opportunistic fungal pathogens that transition from benign commensals to invasive organisms when host immunity weakens or microbiota balance is disrupted. They cause a wide range of diseases, from superficial mucosal infections to life-threatening systemic candidiasis. Although the role of Candida in VAP remains uncertain, growing evidence indicates that it can boost bacterial virulence and facilitate co-colonization. Since VAP isolate profiles serve as a proxy for patient colonization, this study seeks to quantify the relationship between Candida and each of the three bacterial pathogens in VAP cohorts. The authors undertake a meta-regression of 84 ICU cohorts (from 67 studies from 1982 to 2020), evaluating the correlation between respiratory-tract isolation of Candida spp. and three key bacterial VAP pathogens (S. aureus, P. aeruginosa, Acinetobacter spp.). The key result is a robust, independent positive association between Candida isolation and both S. aureus and P. aeruginosa, and a more modest one with Acinetobacter. This work contributes to our understanding of microbial interactions in VAP and may suggest future interventional studies targeting fungal colonization to mitigate bacterial pneumonia risk. This work integrates extensive clinical and microbiological data, is well-referenced, and holds significant relevance for both patient care and research.
Comments:
Table 1 The wide variation in cohort sizes and geographic origins indicates substantial heterogeneity; including a summary metric (e.g., I² or between-study variance) would help readers interpret these correlations. Additionally, since some table cells are left blank (likely representing zero counts), the legend should clarify how missing data and true zeros are distinguished.
Table 2 If possible, please quantify how much the incidence of bacterial VAP rises with each increase in the proportion of Candida.
Figures 1-3 Please include the cohort sizes in each figure’s legend. Note that restricting the analysis to studies that explicitly report Candida in VAP isolates could introduce publication bias. Additionally, could you provide the proportion of VAP patients who had P. aeruginosa, Acinetobacter species, or S. aureus isolates without Candida?
Author Response
Comments 1: Candida species—most notably Candida albicans—are opportunistic fungal pathogens that transition from benign commensals to invasive organisms when host immunity weakens or microbiota balance is disrupted. They cause a wide range of diseases, from superficial mucosal infections to life-threatening systemic candidiasis. Although the role of Candida in VAP remains uncertain, growing evidence indicates that it can boost bacterial virulence and facilitate co-colonization. Since VAP isolate profiles serve as a proxy for patient colonization, this study seeks to quantify the relationship between Candida and each of the three bacterial pathogens in VAP cohorts. The authors undertake a meta-regression of 84 ICU cohorts (from 67 studies from 1982 to 2020), evaluating the correlation between respiratory-tract isolation of Candida spp. and three key bacterial VAP pathogens (S. aureus, P. aeruginosa, Acinetobacter spp.). The key result is a robust, independent positive association between Candida isolation and both S. aureus and P. aeruginosa, and a more modest one with Acinetobacter. This work contributes to our understanding of microbial interactions in VAP and may suggest future interventional studies targeting fungal colonization to mitigate bacterial pneumonia risk. This work integrates extensive clinical and microbiological data, is well-referenced, and holds significant relevance for both patient care and research.
Response 1: Many thanks for this comment which requires no response. May I use the last two sentences here to respond to reviewer 1?
Comments:
Comments 2: Table 1 The wide variation in cohort sizes and geographic origins indicates substantial heterogeneity; including a summary metric (e.g., I² or between-study variance) would help readers interpret these correlations. Additionally, since some table cells are left blank (likely representing zero counts), the legend should clarify how missing data and true zeros are distinguished.
Response 2: These are regression not meta-regression models [this is corrected throughout the manuscript], hence there was no I² to report. The table cells that were left blank (representing counts not reported) are now filled with a ‘.’ As explained in the [new] footnote]
Comments 3: Table 2 If possible, please quantify how much the incidence of bacterial VAP rises with each increase in the proportion of Candida.
Response 3: supplied in legends to each figure. Also sentence added at end of 3.3 [line 209-211]
Comments 4: Figures 1-3 Please include the cohort sizes in each figure’s legend. Note that restricting the analysis to studies that explicitly report Candida in VAP isolates could introduce publication bias. Additionally, could you provide the proportion of VAP patients who had P. aeruginosa, Acinetobacter species, or S. aureus isolates without Candida?
Response 4: cohort sizes supplied. Restricting the analysis to studies that explicitly report Candida in VAP isolates could introduce publication bias – this is now added as a limitation. It is not possible to provide the proportion of VAP patients who had P. aeruginosa, Acinetobacter species, or S. aureus isolates without Candida as these studies [without Candida ] were specifically excluded.
Round 2
Reviewer 1 Report
Comments and Suggestions for Authors
No further comments.